# Declining, seasonal-varying emissions of sulfur hexafluoride from the United States

Lei Hu[1*], Deborah Ottinger[2], Stephanie Bogle[2], Stephen A. Montzka[1], Philip L. DeCola[3,4], Ed Dlugokencky[1], Arlyn Andrews[1], Kirk Thoning[1], Colm Sweeney[1], Geoff Dutton[1,5], Lauren Aepli[2], Andrew Crotwell[1,5]

[1] Global Monitoring Laboratory, US National Oceanic and Atmospheric Administration, Boulder, CO, USA
[2] Climate Change Division, US Environmental Protection Agency, Washington D.C., USA
[3] Department of Atmospheric and Oceanic Sciences, University of Maryland, College Park, MD, USA
[4] GIST.Earth LLC, Washington DC, USA
[5] Cooperative Institute for Research in Environmental Sciences, University of Colorado-Boulder, Boulder, CO, USA

*Correspondence to lei.hu@noaa.gov

## Abstract

Sulfur hexafluoride ($SF_6$) is the most potent greenhouse gas and its atmospheric abundance, albeit small, has been increasing rapidly. Although $SF_6$ is used to assess atmospheric transport modeling and its emissions influence the climate for millennia, $SF_6$ emission magnitudes and distributions have substantial uncertainties. In this study, we used NOAA's ground-based and airborne measurements of $SF_6$ to estimate $SF_6$ emissions from the U.S. between 2007 and 2018. Our results suggest a substantial decline of U.S. $SF_6$ emissions, a trend also reported in the U.S. Environmental Protection Agency (EPA)'s national inventory submitted under the United Nations Framework on Climate Change, implying that U.S. mitigation efforts have had some success. However, the magnitudes of annual emissions derived from atmospheric observations are 40 – 250% higher than the EPA national inventory and substantially lower than the Emissions Database for Global Atmospheric Research inventory. The regional discrepancies between atmosphere-based estimate and EPA's inventory suggest that emissions from electric power transmission and distribution (ETD) facilities and an $SF_6$ production plant that did not or does not report to EPA may be underestimated in the national inventory. Furthermore, the atmosphere-based estimates show higher winter than summer emissions of $SF_6$. These enhanced wintertime emissions may result from increased maintenance of ETD equipment in southern states and increased leakage through aging brittle seals in ETD in northern states in winter. The results of this study demonstrate the success of past U.S. $SF_6$ emission mitigations, and suggest substantial additional emission reductions might be achieved through efforts to minimize emissions during servicing or through improving sealing materials in ETD.

## Short Summary

Effective mitigation of greenhouse gas (GHG) emissions relies on an accurate understanding of emissions. Here we demonstrate the added value of using inventory- and atmosphere-based approaches for estimating U.S. emissions of $SF_6$, the most potent GHG known. The

**results suggest a large decline in U.S. SF₆ emissions, shed light on the possible processes**
**causing the differences between the independent estimates, and identify opportunities for**
**substantial additional emission reductions.**

## Introduction

Sulfur hexafluoride ($SF_6$) is the greenhouse gas (GHG) with the largest known 100-year global
warming potential (GWP) (i.e., 25,200) and an atmospheric lifetime of 580 - 3200 years (Forster
et al., 2021; Ray et al., 2017). $SF_6$ is primarily used in electrical circuit breakers and high-voltage
gas-insulated switchgear in electric power transmission and distribution (ETD) equipment, and its
emissions occur during manufacturing, use, servicing, and disposal of the equipment. There is
also usage and associated emissions of $SF_6$ from production of magnesium and
electronics. Because of its extremely large GWP and long atmospheric lifetime, emissions of $SF_6$
accumulate in the atmosphere and will influence Earth's climate for thousands of years. Since
1978, global emissions of $SF_6$ have increased by a factor of 4 due to rapid expansion of the ETD
systems and the metal and electronics industries (Rigby et al., 2010; Simmonds et al., 2020). As
a result, the global atmospheric mole fractions and radiative forcing of $SF_6$ have increased by 14
times over the same period. In 2019, the radiative forcing of $SF_6$ was 6 mW m$^{-2}$ or 0.2% of total
radiative forcing from all long-lived GHGs, making it the 11$^{th}$ largest contributor to the total
radiative forcing among all the long-lived greenhouse gases and the 7$^{th}$ largest contributor among
gases whose atmospheric concentrations are still growing (i.e., other than CFCs and carbon
tetrachloride) (Gulev et al., 2021). If global $SF_6$ emissions continue at the 2018 rate (9 Gg yr$^{-1}$)
into the future, the global atmospheric mole fraction and radiative forcing of $SF_6$ will linearly
increase by another factor of 4 by the end of the 21$^{st}$ century (Fig. S1). If global emissions of $SF_6$
continue to rise at the same rate as 2000 – 2018, the global atmospheric mole fraction and radiative
forcing of $SF_6$ will increase by another factor of 10 by the end of the 21$^{st}$ century (Fig. S1).
Consistent with the large GWP of $SF_6$ emissions and its importance for influencing climate for
many years, national emissions of this gas are reported under the United Nations Framework
Convention on Climate Change (UNFCCC) annually by the U.S. Furthermore, accurate estimates
of the magnitude and distribution of $SF_6$ emissions are also important in studies to refine our
understanding of atmospheric transport processes in the troposphere and stratosphere (Orbe et al.,
2021; Waugh et al., 2013; Denning et al., 1999; Gloor et al., 2007; Peters et al., 2004; Schuh et al.,
2019; Ray et al., 2017; Maiss and Levin, 1994; Harnisch et al., 1996).
Although global emissions of $SF_6$ can be well constrained with knowledge of its observed remote-
atmosphere growth rates and its atmospheric lifetime, large uncertainties remain in the magnitude
and distribution of $SF_6$ emissions on national and regional scales. For example, the total annual
national emissions reported to the UNFCCC summed from its Annex I (mostly developed
countries) and some non-Annex I (mostly developing) countries (including China, one of the large
$SF_6$ emitting countries) account for only 50% of global annual $SF_6$ emissions derived from
atmospheric observations for the analyzed years (1990 – 2007) (Simmonds et al., 2020; Rigby et
al., 2010; Levin et al., 2010). This difference between activity-based inventory ("bottom-up")
estimates and atmosphere-based ("top-down") estimates may result from underestimates of
emissions by activity-based inventories (Simmonds et al., 2020; Rigby et al., 2010; Levin et al.,
2010; Weiss and Prinn, 2011) as well as from substantial emissions from non-reporting countries.
The results of activity-based inventories are sensitive to estimated activity levels and, especially,
emission rates. In the Emission Database for Global Atmospheric Research (Janssens-Maenhout,
2011; Crippa et al., 2020), U.S. $SF_6$ emissions were up to five times larger than the emissions
estimated by the U.S. Environmental Protection Agency (EPA) and in their reporting to the
UNFCCC (U.S. Environmental Protection Agency, 2022a) (Fig. 1). The causes for this large
difference are not fully known, but appear to arise largely from the ETD sector (Fig. S2).
Uncertainties in EPA's emissions estimates were also illuminated by a comparison between the
$SF_6$ usage inferred from the user reports (which form the basis of EPA's emissions estimates) and
the $SF_6$ usage inferred from suppliers' reports, which showed that supplier-based estimates were
70% higher than user-based estimates in 2012 (Ottinger et al., 2015).

Against this backdrop, we estimated U.S. $SF_6$ emissions between 2007 and 2018 using inverse
modeling of atmospheric mole fraction measurements made from ground-based and airborne
whole-air flask samples collected from the U.S. National Oceanic and Atmospheric Administration
(NOAA) Global Greenhouse Gas Reference Network (Fig. 1). The analysis provides robust
emission estimates by region and season for the contiguous U.S. (CONUS). Our study offers an
independent estimate that complements the current U.S. inventory-based national emission
reporting of $SF_6$ to the UNFCCC. This effort exemplifies the quality assurance guidance laid out
in the *2019 Refinement to the 2006 IPCC Guidelines for National Greenhouse Gas Inventories*,
which states "Atmospheric measurements are being used to provide useful quality assurance of the
national greenhouse gas emission estimates. Under the right measurement and modelling
conditions, they can provide a perspective on the trends and magnitude of greenhouse gas (GHG)
emission estimates that is largely independent of inventories" (Maksyutov et al., 2019). In fact,
the United Kingdom, Switzerland, and Australia have already included top-down atmosphere-
based emission estimates in the QA/QC section of their national GHG emission reporting to
UNFCCC (Fraser et al., 2014; Henne et al., 2016; Manning et al., 2021). The United States also
started to include top-down estimates of four major hydrofluorocarbons (HFCs) as a comparison
to the U.S. national GHG inventory reporting in 2022 (U.S. Environmental Protection Agency,
2022a). Derived national and regional $SF_6$ emissions from this analysis are accessible through
NOAA's U.S. Emission Tracker for Potent GHGs website
(https://gml.noaa.gov/hats/US_emissiontracker).


## 123    **Methods**

Top-down atmosphere-based $SF_6$ emission estimates were derived using inverse modeling of
NOAA's long-term atmospheric measurements of $SF_6$
(https://gml.noaa.gov/aftp/data/hats/sf6/Data_in_Hu_et_al_2023/). Measurements made over
North America were based on air samples collected by discrete flasks from tall towers and aircraft.
The tall tower flask samples were typically collected every one to two days and airborne flask-
sample profiles were collected once or twice per month between 0 and 8 km above sea level.
Measurements made outside North America were from weekly whole-air samples collected
globally, generally at remote locations far away from emission sources
(https://gml.noaa.gov/dv/site/). All the whole-air flask samples were shipped to Boulder and
analyzed by a Gas Chromatography with Electron Capture Detector (GC-ECD) for $SF_6$.
Uncertainty of each $SF_6$ flask measurement is approximately 0.04 to 0.05 ppt, which includes
uncertainties related to short-term measurement noise, long-term measurement reproducibility,
and calibration scale transfer from gravimetric standards to working standards.

Mole fraction enhancements of $SF_6$ over the contiguous U.S. (Fig. 1) relative to $SF_6$ mole fractions
in air measured upwind were then estimated for deriving U.S. emissions. These enhancements
were estimated by referencing them to "background" mole fractions that were derived using three
different approaches.  These approaches are similar to previous inversion analyses for other
atmospheric trace gases (Hu et al., 2021; Hu et al., 2017).  In all three approaches, we constructed
an empirical 4-dimensional mole fraction field based on measurements made in air over the Pacific
and Atlantic Ocean basins and in the free troposphere above 3 km over North America, so that it
contains vertical and horizontal gradients of mole fractions measured in the remote atmosphere
over time.  From this empirical background, we then extracted the mole fraction at the sampling
time and location of each observation and used it as our first background estimate.  In the second
approach, we considered 500 air back-trajectories associated with each observation.  Five hundred
background estimates were extracted from this empirical background at the locations where the air
back-trajectories exited the North American domain horizontally or where they were aloft above
5 km.  In most cases, the majority of particles exited North America horizontally or vertically
within 10 days, but for those that remained within the domain after 10 days, background values
were derived from their positions 10 days after sampling.  For midcontinent and eastern sites, there
were up to 20% of particles remained within the domain after 10 days. The 500 background
estimates were averaged to obtain the background mole fraction estimation for that observation.
In the third approach, we assessed potential biases in the background estimate from the second
approach, particularly because there was a small fraction of back-trajectories ending up in the
planetary boundary layer in North America after 10 days.  Background mole fractions for these
particles were likely higher than estimated using the marine boundary layer information.  To
minimize such biases, we corrected our background estimates from the second approach based on
their differences with measurements made within North America that had small surface
sensitivities over populated areas (Hu et al., 2017).
$SF_6$ mole fraction enhancements estimated in observations at North American sites were then
incorporated into a regional inverse model to estimate U.S. national and regional emissions,
following the same methodology as described in our previous inversion studies for other
anthropogenic gases (Hu et al., 2017; Hu et al., 2015; Hu et al., 2016).  In regional inversions, we
assume a linear relationship between atmospheric mole fraction enhancements and upwind
emissions.  The linear operator is called a 'footprint' or the Jacobian matrix, representing the
spatial and temporal sensitivity of atmospheric mole fraction observations to emissions.  Footprints
were computed by two transport models, the coupled Weather Research and Forecasting -
Stochastic Time-Inverted Lagrangian Transport model (WRF-STILT) (Nehrkorn et al., 2010) and
the Hybrid Single-Particle Lagrangian Integrated Trajectory model (Stein et al., 2015) driven by
the North American Mesoscale Forecast System (HYSPLIT - NAMs).  The WRF field has 41
pressure levels and a horizontal resolution of 10-km in North America and 40-km outside of North
America.  The NAMs meteorology has a 12-km resolution and 40 sigma-pressure levels. Before
March 2009, when NAMs was not available; we used NAM-12 meteorology, which only has 26
vertical levels.  NAMs or NAM-12 was nested with the U.S. National Centers for Environmental
Prediction (NCEP) 0.5° Global Data Assimilation System (GDAS0.5) with 55 sigma-pressure
levels.  Both WRF-STILT and HYSPLIT-NAMs were run with 500 particles back in time for 10
days (e.g., Miller et al., 2013; Nevison et al., 2018; Miller et al., 2012; Gerbig et al., 2003).  In
each run, particles were released at the sampling inlet heights.  Footprints were then calculated by
integrating particles between the modeled surface to modeled boundary layer in each grid at each
timestep (Lin et al., 2003).
A Bayesian inverse modeling technique (Rodgers, 2000) was implemented, where a prior emission
field or "a priori" was required.  The model adjusts magnitudes and distributions of the a priori at
a $1^o \times 1^o \times$ weekly resolution, such that the posterior solution of emissions better represents the
observed magnitudes, and horizontal and vertical gradients of mole fraction enhancements
observed in the U.S.  Here, we used two different temporally-constant prior emission fields.  The
first one was from the Emissions Database for Global Atmospheric Research version 4.2
(EDGARv4.2) with a U.S. total $SF_6$ emission of 1.8 Gg yr$^{-1}$ in 2008,  We used the 2008
EDGARv4.2 estimate and applied it for all years between 2007 – 2018 in our inversions.
EDGARv4.2 was the most recent grid-scale product offered by EDGAR at the time we conducted
our inversions.  EDGAR version 7.0 (EDGARv7.0) became available only after this work was
submitted in late September of 2022.  It extends this inventory emission through 2021 and its U.S.
total and regional $SF_6$ emissions for earlier years are similar to those in EDGARv4.2 (Figs. 1 and
2).  Given the similarities of EDGAR v7.0 with v4.2 in distribution and magnitude and the
insensitive nature of our posterior results to these aspects of the prior (see below), we did not rerun
inversions with EDGARv7.0 as a priori.  The second a priori includes a U.S. total emission of 0.4
Gg yr$^{-1}$ for 2007 – 2018.  It was distributed by population density from the Gridded Population of
the World (GPW) v4 dataset (https://sedac.ciesin.columbia.edu/data/collection/gpw-v4, last
access: 15 March 2019).  The weight between the prescribed prior emissions and atmospheric
observations in the final posterior emission solution was determined by the values in the prior
emission error covariance matrix and the model-observation mismatch covariance matrix, which
were calculated from the maximum likelihood estimation (Michalak et al., 2005; Hu et al., 2015).
In each inversion, the derived $1^o \times 1^o \times$ weekly emissions and emission uncertainties were
aggregated to derive emissions and uncertainties at regional and national scales and at monthly
and annual time steps.  When calculating the posterior uncertainty, we considered the temporal
and spatial correlations of posterior errors in the derived full posterior emission covariance matrix.
The final reported emissions and emission uncertainties include results from a total of 12
inversions that have two representations of transport, two prior emission fields, and three
background estimates.  Assume $\mu_i$ and $\sigma_i$ represent the posterior emission estimate and its
associated 1-sigma error for the $i$ th inversion.  Our final estimate of emissions and its associated
uncertainty discussed in the text were calculated as the mean posterior emission and the 2-sigma
uncertainty $(2\sigma_t)$ derived from Eq. (1).
$$\sigma_t = \sqrt{\frac{\sigma_1{}^2 + \sigma_2{}^2 + \cdots + \sigma_{12}{}^2}{12} + \sigma_s{}^2} \qquad (1)$$

where $\sigma_s$ denotes 1-sigma spread or variability of the posterior emissions derived from all 12
inversions.

## Results and discussion
### Declining $SF_6$ emissions from the U.S.
The U.S. recognized that it had significant emissions of $SF_6$ in the 1990s and has taken steps to
reduce its national emissions.  In the U.S., 60 – 80% of $SF_6$ emissions have historically been from
the ETD sector (U.S. Environmental Protection Agency, 2022a) (Fig. 1).  Outside the ETD sector,
smaller amounts of $SF_6$ are used in semiconductor manufacturing processes as a source of fluorine
to etch patterns onto chips and to clean thin film deposition chambers, and $SF_6$ is also used as a
cover gas in magnesium production and casting processes to prevent rapid oxidation of molten
magnesium.  Both of these uses result in emissions. $SF_6$ emissions from magnesium processes
accounted for roughly $15 - 30$ % of the U.S. total emissions reported by EPA between 1990 and
2018 (Fig. 1).  While the magnitude of $SF_6$ emissions from the electronics manufacturing sector
has not changed much over time, its share of total U.S. $SF_6$ emissions in the EPA inventory has
increased from 2% in 1990 to 14% in 2018 as emissions from other industries have decreased.
Since 1999, the U.S. EPA has worked with the electric power industry through the voluntary $SF_6$
Emission Reduction Partnership for Electric Power Systems to identify, recommend, and
implement cost-effective solutions to reduce $SF_6$ emissions.  There have also been regulations at
the state level to reduce $SF_6$ emissions from the ETD sector (U.S. Environmental Protection
Agency, 2022a).  In addition, the EPA operated voluntary partnership programs with the
semiconductor and magnesium industries from the late 1990s through 2010 to understand and
reduce their emissions. These national- and state- level mitigation strategies, along with an increase
in the market price of $SF_6$ during the 1990s, have resulted in a substantial reduction in total U.S.
$SF_6$ emissions since 1990  (Fig. 1).  In addition, before 2011, $SF_6$ was likely emitted from an $SF_6$
production plant that ceased producing $SF_6$ in 2010, according to data reported to the U.S. EPA.
Total U.S. $SF_6$ emissions estimated by the EDGAR version 4.2 or 7.0 inventory showed an
absolute decline over this period similar to that in the EPA National Greenhouse Gas Inventory
(GHGI), but EDGAR emissions were substantially larger on average (Fig. 1).  Note that although
the U.S. national total in EDGARv7.0 suggests lower emissions than EDGARv4.2, this difference
arises only in the magnesium production sector.  There were slightly higher emissions from the
ETD and electronics industries in EDGARv7.0 than in EDGARv4.2 over the U.S. (Fig. S2).
Consistent with the inventory reports, the independent, atmospheric observation-based results
presented here suggest a large decline of U.S. total $SF_6$ emissions, confirming the success of U.S.
$SF_6$ emission mitigation efforts.  The atmospheric observation-based emissions declined from 0.93
($\pm 0.19, 2\sigma$) Gg yr$^{-1}$ in $2007 - 2008$ to 0.37 ($\pm 0.10, 2\sigma$) Gg yr$^{-1}$ in $2017 - 2018$ (Table 1 and Fig.
1).  The $0.56 \pm 0.21$ ($2\sigma$) Gg yr$^{-1}$ drop in $SF_6$ emissions from $2007 - 2008$ to $2017 - 2018$ is
equivalent to a reduction of $13 \pm 7$ million tons of $CO_2$ emission, when using the 100-year global
warming potential that was used in the EPA GHGI ($GWP_{100} = 22800$).
Although both the atmosphere-based top-down and inventory-based bottom-up estimates show
declining trends for total U.S. $SF_6$ emissions, the estimated emission magnitudes are quite different.
In $2007 - 2008$, the atmosphere-based emissions fall between the EDGARv7.0 and EPA's GHGI
estimates; but the difference between the atmosphere-based estimate and EDGARv7.0 increases
over time, whereas the difference between the atmosphere-derived estimate and EPA's inventory
decreases over time.  After 2011, the atmosphere-based emission estimates are 0.93 ($\pm 0.07, 2\sigma$ )
Gg yr$^{-1}$ (a factor of 3.4) lower than EDGARv7.0 and only about 0.15 ($\pm 0.07, 2\sigma$) Gg yr$^{-1}$ (35%)
higher than the EPA's GHGI (U.S. Environmental Protection Agency, 2022a).  The improved
agreement between the EPA GHGI and the atmosphere-based estimates may be associated with
more accurate emission information used to inform the EPA's GHGI after 2010.  Before 2011, the
$SF_6$ emission estimate in the EPA GHGI was primarily informed by reporting through the
voluntary Partnership Programs between EPA and various industries (Rand, 2012) described above.
In 2011, EPA established its greenhouse gas reporting program (GHGRP), requiring facility-based
reporting of greenhouse gas (GHG) data and other relevant information from large GHG emission
sources ($\geq$ 25,000 $CO_2$-equivalent metric tons of GHG emissions per year). Although smaller
emitters are not required to report their emissions, this program provides more complete emission
information than had been available prior to 2011. For example, from 1999 to 2010, ETD facilities
representing an estimated 60% of the emitting activity reported their $SF_6$ emissions to EPA through
EPA's voluntary reporting program. After 2010, ETD facilities representing an estimated 70% of
the emitting activity began reporting their emissions to EPA under the GHGRP.
A variety of factors may be contributing to the difference observed between the $SF_6$ emissions
estimates from atmospheric measurements and the estimates developed for the U.S. EPA GHGI.
The largest potential contributor to the difference is a possible underestimate by the GHGI of
emissions from ETD facilities that do not report to EPA, or that did not report to EPA until they
were required to report by the GHGRP starting in 2011. Emissions from non-reporting facilities
are currently estimated based on the uncertain assumption that the emission rate per mile of
transmission line (transmission mile) for non-reporting facilities has been the same, on average, as
that for reporting facilities in each year of the time series. However, the emission rate per
transmission mile has varied significantly across facilities and over time due to a variety of factors,
including the age of the electrical equipment, maintenance practices, local regulations, and the
quantity of $SF_6$-containing equipment per transmission mile ($SF_6$ nameplate capacity). Among
reporting facilities, the emission rate has fallen from an average of 0.7 kg per transmission mile in
1999 to 0.2 kg per transmission mile in recent years, with emission rates declining most quickly
in the first three years of reporting (i.e., 1999-2001 for Partners, 2011-2013 for facilities that began
reporting under the GHGRP). This implies that reporting itself may drive emission reductions.
Thus, it is plausible that the emission rate of non-reporting facilities has fallen more slowly than
that of reporting facilities.
In the years prior to 2011, there are several additional factors that may be contributing to the
underestimate of $SF_6$ emissions by the GHGI, compared to the atmosphere-based estimates. One
potentially significant factor is that the GHGI does not currently account for $SF_6$ emissions from
the $SF_6$ production plant that operated in Metropolis, Illinois, up until 2010. This plant never
reported its emissions to EPA; but based on production capacity data for the plant from 2006 and
the broad range of emission factors observed for production of $SF_6$ and other fluorinated gases, the
plant's $SF_6$ emissions would likely have ranged between 0.03 and 0.3 Gg yr$^{-1}$. Notably, the region
showing the largest drop in the atmosphere-derived emissions between 2008 and 2011-2018
includes Metropolis, Illinois (Fig. S3). Although emissions from this plant have not been included
in previous GHGIs, the discrepancy highlighted here points to potential significant contributions
from this plant before 2011 (and other fluorinated gas production facilities) that will be included
in future submissions of the GHGI.
Other factors that may account for a small portion of the post-2011 difference is an underestimate
of emissions of $SF_6$ from electronics manufacturing by a factor of 2 (equivalent to $\sim$ 0.02 Gg yr$^{-1}$).
In the GHGI, the EPA adjusted the time series of GHGRP-reported data for 2011 through 2013 to
ensure time-series consistency using a series of calculations that took into account the
characteristics of a facility (e.g., wafer size and abatement use) and updated default emission
factors and destruction and removal efficiencies. These updates reflected improved activity data
and not changes to emission rates, and resulted in an increase in $SF_6$ emissions estimates by 95%
from electronics manufacturing. Finally, a similar improvement for time series consistency is
planned for pre-2011 estimates and is expected to result in a similar relative increase in estimated
$SF_6$ emissions from the electronics sector for those years.
**U.S. regional $SF_6$ emissions**
We also investigated regional emissions derived from atmospheric inversions and from EPA's
recently created Inventory of U.S. Greenhouse Gas Emission and Sinks by State (U.S.
Environmental Protection Agency, 2022b) to understand the distribution of $SF_6$ emissions and how
various regions contribute to the difference between the atmosphere- and inventory- derived U.S.
total emissions. Note that the EPA GHGI was only able to allocate 20 – 30 % of ETD emissions
to a single state by facility location (i.e. when the facility was only in one state). The remaining
emission was distributed based on a national average emission factor (kg of $SF_6$ per transmission
mile). Because of this limited regional resolution, we expect some limitations in the regional
estimates of the GHGI. However, this comparison with atmosphere-based estimates helps assess
the robustness of the regional estimates.
The atmosphere-based emission estimates suggest that about 80% of the U.S. total $SF_6$ emissions
were contributed by three regions: the northeast, central north, and central south (Table 1; Figs. 2).
Regional $SF_6$ emissions corresponding to the GHGI calculated using the EPA's Inventory of U.S.
Greenhouse Gas Emission and Sinks by State (U.S. Environmental Protection Agency, 2022b)
were distributed slightly differently. For the southeast, west, and mountain regions, EPA's
regional emissions agree well with emissions estimated from atmospheric observations, but they
are lower than the atmosphere-derived emissions in the northeast for the entire study period and in
the central north and central south during 2007 – 2010. Such regional differences were expected
due to the limited regional resolution of the GHGI for emissions from ETD. For regions that
predominantly had emissions from the ETD sector, the difference is likely more dependent on how
similar the ETD emissions in the region are to the national average. This method could result in
an underestimate of emissions in the regions like the northeast where the average emission rate is
expected to be higher than the national average based on historical data submitted to the EPA by
facilities in the region. Higher regional emission rates in the northeast could be due in part to the
region containing more gas-insulated equipment per transmission mile and the presence of older
transmission systems (i.e. older, leakier equipment). The national average emission factor may be
more appropriate for the mountain, central north, and central south regions. This is because
regional emission factors that are based only on GHGRP reported emissions from facilities that
reside entirely within the region, are similar to a national average in these regions. Better
agreement in the western region may be also associated with the incorporation of the California
Air Resource Board estimate for $SF_6$ from California in the GHGI.
For the central north and central south regions, the atmosphere-derived emissions were higher in
2007 – 2010 and show a larger declining trend than the EPA GHGI. The larger discrepancy in the
central north and central south before 2011 may be due in part to the unaccounted emissions by
GHGI from the $SF_6$ production facility in Metropolis, Illinois, described above, which ceased
producing $SF_6$ in 2010. This facility is located right at the border between the central north and
central south regions, so it is likely that emissions from it could have been attributed to one or both
adjacent regions in the atmospheric inversions.
Besides the EPA GHGI, we also compared our top-down estimate with the EDGAR inventories
(EDGARv4.2 and EDGARv7.0). Results suggest that emissions estimated by the EDGAR are
higher than the atmosphere-derived emissions and the EPA's inventory estimates for all regions
across the U.S., especially in the western and northeast regions (Fig. 2).
**Significant seasonality detected in U.S. $SF_6$ emissions**
The monthly $SF_6$ emissions derived from our inverse analysis of atmospheric concentration
measurements reveal a prominent seasonal cycle with higher emissions in winter for all 12 years
of this analysis (Fig. 3a). On average, the magnitude of winter $SF_6$ emissions is about a factor of
2 larger than summer emissions summed across the contiguous U.S. (Fig. 3a). This seasonality is
most likely from the use, servicing, and disposal of ETD equipment, as $SF_6$ emissions from
magnesium production, electronics production, and manufacturing of ETD equipment are
expected to be aseasonal. Consistent with this hypothesis, winter-to-summer ratios of total U.S.
$SF_6$ emissions derived for individual years significantly correlate at a 99% confidence level ($r = 
0.71$; $P = 0.01$) with the annual fractions of national emissions contributed by the ETD sector
reported by EPA (Fig. 3c). Moreover, this robust relationship also holds regionally ($r = 0.84$; $P = 
0.02$) (Fig. 3c). The largest seasonal variation in emissions is detected in the southeast and central
south regions of the U.S., where the ETD sector accounted for more than 85% of the regional total
emissions (Figs. 2, 3b, and 3c). In these southern regions, the winter emissions were higher than
summer emissions by more than a factor of 2, whereas in the central north, where the ETD sector
accounted for about 50% of the regional total emissions, the mean winter-to-summer emission
ratio was less than 1.5 (Figs. 2 and 3c).
The enhanced winter emissions in the southern states are consistent with the fact that more
servicing is performed on electrical equipment and transmission lines over this region in the cooler
months (information provided by Mr. B. Lao at the DILO Company, Inc.), when electricity usage
is lower compared to other seasons (U.S. Energy Information Administration, 2020). This
suggests that the enhanced seasonal $SF_6$ emission may be associated with the season during which
electrical equipment repair and servicing is enhanced. In the northern states, the higher winter
than summer emissions may relate to increased leakage through more brittle seals in the aging
electrical transmission equipment due to increased thermal contraction in winter (Du et al., 2020).
This winter-to-summer ratio in the northeast is somewhat higher than in the other northern regions
(Fig. 3b), which may reflect the fact that the electrical power grid is denser (U.S. Federal
Emergency Management Agency, 2008) and ETD is the primary emitting source of $SF_6$ over the
northeast region.
Given that the ETD sector may be the primary cause for seasonally-varying emissions in the U.S.,
we next assessed changes in seasonality over time and their implication for changes in sector-
based emissions from 2007 to 2018. The most notable feature of the time series (Fig. S4) is that
the largest seasonal cycle occurred in 2009 when the economic recession took place. The 2009
recession resulted in a significant drop in the production of magnesium and electronics (U.S.
Environmental Protection Agency, 2021), but little (if any) change to the ETD infrastructure and
associated servicing practices is likely to have occurred. Thus, ETD emissions represent a larger
fraction of the total U.S. $SF_6$ emissions in that year. In addition, the winter-to-summer emission
ratios appear smaller before the 2009 peak (i.e., in 2007 – 2008) than after it (in 2011 – 2018).
This may imply that emissions from the ETD sector accounted for a growing fraction of total
emissions through this entire period.

**Conclusions and implications**

$SF_6$ is a potent industrially-produced greenhouse gas with an extremely long atmospheric lifetime.
It is a trace gas that is primarily used in the electrification of the energy sector.  In the past five
decades, global emissions, concentrations, and radiative forcing of $SF_6$ have substantially
increased due to growing energy demand.  Without effective emission mitigation efforts
worldwide, the climate impact of $SF_6$ will continue to rise in the future.  In contrast to the global
emission trend, U.S. $SF_6$ emissions have decreased substantially since the 1990s.  These decreases
are documented in EPA's emission inventories reported annually to the UNFCCC and in the new
results reported here from an inverse analysis of atmosphere concentration measurements. These
independently-derived U.S. emission records demonstrate substantial success by U.S. industry in
coordination with the EPA in mitigating $SF_6$ emissions.

The magnitude of $SF_6$ emissions derived from atmospheric inversions are higher than those
reported in the EPA GHGI but lower than EDGAR; but the difference between the EPA GHGI
and atmosphere-derived estimates become substantially smaller after 2011, when national GHG
reporting became mandatory, implying that that the shift from voluntary to mandatory emission
reporting by industry increased the accuracy of the inventory. However, differences remain
between the emissions estimated from these independent methods, which may relate to the
uncertain assumptions about ETD-related emission rates per mile from non-reporting facilities in
the GHGI.  Although the EPA GHGI may underestimate $SF_6$ emissions, its contribution to the
global "missing" source of $SF_6$ is small. More specifically, the total $SF_6$ emissions summed from
all reporting countries to the UNFCCC are only half of the global emissions derived from global-
scale observed concentration trends; in other words, there are ~ 4 Gg $SF_6$ yr$^{-1}$ or 100 million tons
of $CO_2$-equivalent per year of $SF_6$ emissions still "missing" in the global GHG accounting system.
The underestimate of the U.S. GHGI only contributed 14% in 2007 – 2008 and only 3% after 2011
to this global $SF_6$ emission gap, implying either large underreporting of $SF_6$ emissions from other
reporting countries or large emissions from non-reporting countries.

Regional emissions from atmospheric inversions were compared with the recently available
disaggregation of the EPA GHGI by state to provide an initial assessment on the emission
distribution of $SF_6$ estimated from the GHGI.  Good agreement was noted in some regions but not
others.  Combining the spatial discrepancies with processes used for constructing the GHGI, we
were able to identify regions where applying a national average emission factor may be
inappropriate and where historical emissions of a facility (the $SF_6$ production plant in Metropolis,
Illinois) are currently not accounted for but may have been significant.

Finally, the atmosphere-derived results further suggest a strong seasonal cycle in U.S. $SF_6$
emissions from electric power transmission and distribution for the first time, with wintertime
emissions twice as large as summertime emissions.  This seasonal cycle is thought to be strongest
in southern states, where servicing of ETD equipment is typically performed in winter.  The
seasonal cycle is likely enhanced additionally by increased leakage from ETD equipment during
the winter, when cold weather makes sealing materials more brittle and therefore less effective.
This newly discovered seasonal emission variation implies that further larger reductions of $SF_6$
emission in the U.S. might be achievable through efforts to minimize losses during equipment
maintenance and repairs, and through the use of improved sealing materials in ETD equipment.
The 2019 Refinements to the 2006 IPCC Guidelines for National Greenhouse Gas Inventories
suggest that atmospheric inversion-derived emissions be considered in the quality assurance,
quality control and verification of the national GHG inventory reporting. It is anticipated that the
consideration of an independent estimate will lead to more accurate inventories. The work
presented here, however, suggests that a collaboration between these communities can provide
much more. In the case of $SF_6$, the result has been not only an improved understanding of emission
magnitudes, but also a better grasp of the processes that lead to emissions and the identification of
substantial new emission mitigation opportunities, thereby pointing the way towards a more
effective and efficient means to minimize and reduce national greenhouse gas emissions.

**Data availability**
Atmospheric $SF_6$ observations used in this analysis are available at
https://gml.noaa.gov/ccgg/obspack/data.php. Data included in our inversion can be downloaded
at https://gml.noaa.gov/aftp/data/hats/sf6/Data_in_Hu_et_al_2023/. The marine boundary layer
reference for $SF_6$ can be downloaded from https://gml.noaa.gov/ccgg/mbl/data.php. Atmospheric
observation-derived U.S. national and regional emissions from this analysis are accessible through
the US Emission Tracker for Potent GHGs (https://gml.noaa.gov/hats/US_emissiontracker). $SF_6$
emissions reported to the GHGR are available at https://www.epa.gov/enviro/greenhouse-gas-
customized-search.

**Author contributions**
LH developed the inversion framework, performed the analysis, and wrote the paper with DO, SB,
and SM. Significant edits and inputs were also from PD and ED. DO and SB made and provided
EPA $SF_6$ emission estimates. PD, SM, and LH initiated this project. LH, DO, SB, SM, and PD
worked on the interpretation of the results. ED led the NOAA $SF_6$ measurements. PD coordinated
the discussion between NOAA and EPA colleagues. AA led the NOAA tower sampling network,
provided the 4D empirical background estimates of $SF_6$ and WRF-STILT footprints. KT and GD
helped with improving the inversion code. KT and LH computed the HYSPLIT footprints. CS
led the NOAA aircraft sampling network. LA contributed the construction of EPA $SF_6$ emission
estimates. AC contributed to NOAA $SF_6$ measurements.
The views expressed in this article are those of the authors and do not necessarily represent the
views or policies of the U.S. Environmental Protection Agency.

**Acknowledgement**
This work was funded in part by the Grantham Foundation, NOAA Climate Program Office's
Atmospheric Chemistry, Carbon Cycle, and Climate (AC4) and Climate Observations and
Monitoring (COM) programs (NA21OAR4310233), and the
NOAA Cooperative Agreement with CIRES, NA17OAR4320101. We thank Mr. Billy Lao at the
DILO company, Inc. for insights on the timing of repairing and servicing of electrical equipment.
We thank Dr. Bradley Hall for maintaining the primary SF$_6$ calibration scale for all NOAA SF$_6$
measurements. We thank Jon Kofler, Kathryn McKain, Don Neff, Sonja Wolter, Jack Higgs,
Molly Crotwell, Patrick Lang, Eric Moglia, and Monica Madronich for facilitating flask sampling
and measurements.  We thank Dr. Andy Jacobson for providing SF$_6$ simulations from the SF$_6$
Model Inter-comparison Project even though they were not included in the final publication of this
analysis.

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

**Table 1**. U.S. national and regional annual emissions of $SF_6$ (in Gg yr$^{-1}$) reported by EPA and
derived from NOAA atmospheric measurements from this study. Errors derived from NOAA
atmospheric measurements are expressed at a 95% confidence interval.

| Year | National totals | | Regions | | | | | | | | | | | |
| --- | --- | --- | --- | --- | --- | --- | --- | --- | --- | --- | --- | --- | --- | --- |
| | | | Northeast | | Southeast | | Central North | | Central South | | Mountain | | West | |
| | EPA | NOAA | EPA | NOAA | EPA | NOAA | EPA | NOAA | EPA | NOAA | EPA | NOAA | EPA | NOAA |
| 2007 | 0.40 | 0.83±0.19 | 0.04 | 0.18±0.10 | 0.03 | 0.05±0.07 | 0.16 | 0.30±0.07 | 0.06 | 0.22±0.07 | 0.07 | 0.04±0.03 | 0.04 | 0.03±0.07 |
| 2008 | 0.37 | 1.03±0.26 | 0.05 | 0.22±0.10 | 0.02 | 0.09±0.07 | 0.13 | 0.34±0.14 | 0.06 | 0.28±0.09 | 0.06 | 0.06±0.04 | 0.03 | 0.04±0.06 |
| 2009 | 0.32 | 0.75±0.26 | 0.04 | 0.16±0.10 | 0.03 | 0.08±0.05 | 0.10 | 0.25±0.13 | 0.06 | 0.22±0.10 | 0.06 | 0.02±0.03 | 0.03 | 0.02±0.04 |
| 2010 | 0.32 | 0.63±0.16 | 0.04 | 0.12±0.04 | 0.02 | 0.05±0.03 | 0.11 | 0.20±0.08 | 0.06 | 0.12±0.04 | 0.06 | 0.05±0.02 | 0.03 | 0.08±0.04 |
| 2011 | 0.36 | 0.58±0.12 | 0.04 | 0.13±0.04 | 0.03 | 0.06±0.02 | 0.14 | 0.19±0.05 | 0.06 | 0.11±0.03 | 0.06 | 0.03±0.02 | 0.03 | 0.06±0.02 |
| 2012 | 0.30 | 0.40±0.11 | 0.04 | 0.09±0.05 | 0.03 | 0.04±0.03 | 0.10 | 0.13±0.03 | 0.05 | 0.08±0.03 | 0.05 | 0.03±0.02 | 0.03 | 0.04±0.02 |
| 2013 | 0.28 | 0.40±0.12 | 0.04 | 0.11±0.07 | 0.03 | 0.03±0.03 | 0.08 | 0.13±0.04 | 0.04 | 0.08±0.03 | 0.05 | 0.02±0.02 | 0.03 | 0.03±0.02 |
| 2014 | 0.29 | 0.48±0.14 | 0.04 | 0.15±0.09 | 0.03 | 0.05±0.03 | 0.09 | 0.13±0.04 | 0.05 | 0.08±0.03 | 0.05 | 0.03±0.02 | 0.03 | 0.03±0.02 |
| 2015 | 0.24 | 0.43±0.14 | 0.04 | 0.13±0.05 | 0.02 | 0.05±0.03 | 0.08 | 0.13±0.04 | 0.04 | 0.07±0.02 | 0.04 | 0.03±0.02 | 0.02 | 0.03±0.02 |
| 2016 | 0.26 | 0.40±0.09 | 0.04 | 0.12±0.04 | 0.03 | 0.04±0.02 | 0.10 | 0.11±0.03 | 0.04 | 0.08±0.04 | 0.04 | 0.02±0.02 | 0.02 | 0.02±0.02 |
| 2017 | 0.26 | 0.35±0.12 | 0.03 | 0.09±0.04 | 0.03 | 0.03±0.03 | 0.09 | 0.10±0.03 | 0.04 | 0.08±0.04 | 0.04 | 0.03±0.03 | 0.02 | 0.02±0.02 |
| 2018 | 0.25 | 0.39±0.12 | 0.03 | 0.09±0.02 | 0.03 | 0.07±0.04 | 0.09 | 0.11±0.03 | 0.04 | 0.08±0.04 | 0.04 | 0.02±0.02 | 0.02 | 0.03±0.03 |



**Figures**


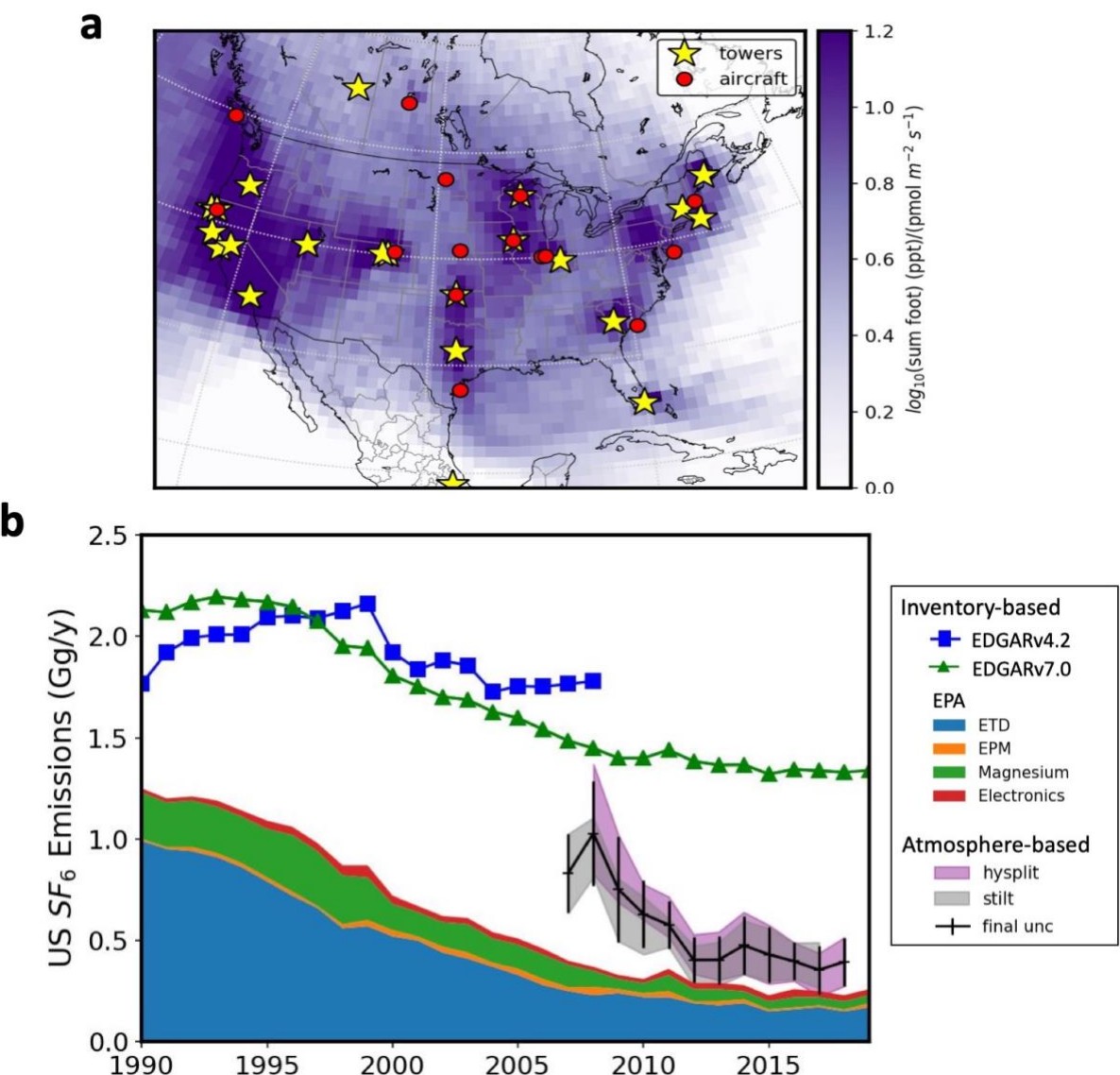

**Fig. 1**. U.S. SF$_6$ emissions derived from atmospheric observations and reported by inventories. (a)
Locations of atmospheric SF$_6$ measurements considered in the regional inversions; tower-based
sampling is indicated as stars and airborne-profile sampling is denoted as circles. Sensitivity of the
atmospheric SF$_6$ measurements to surface emissions is indicated on a log10 scale as purple shading.
(b) U.S. SF$_6$ emissions reported by EDGAR (v4.2 and v7.0) and EPA inventories, and derived
from atmospheric observations. National totals are shown from EDGARv7.0, whereas the EPA
inventory is parsed out by sector, including electric power transmission and distribution (ETD),
electrical equipment manufacturing (EPM), magnesium production, and electronics. Atmosphere-
based emission estimates for the contiguous U.S. are derived with two different model analyses of
the atmospheric observations using two different transport simulations (HYSPLIT-NAMS in
purple shading for 2008 - 2018 and WRF-STILT in gray shading for 2007 - 2017). The black line
with error bars indicates inversion ensemble annual means and an uncertainty at a 95% confidence
interval.

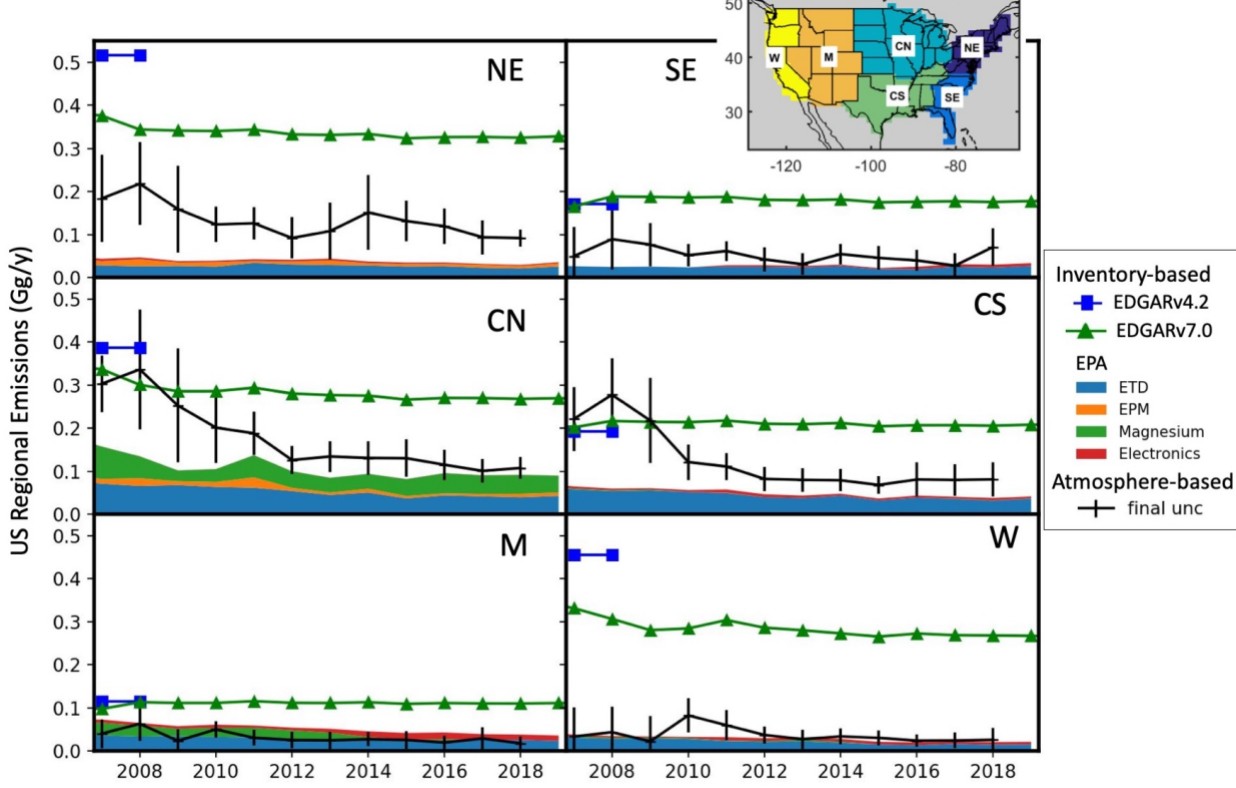

**Fig. 2**. Regional SF$_6$ emissions over the U.S., derived from atmospheric observations and reported
by EPA's GHGI and EDGAR (EDGARv4.2 and EDGARv7.0). EPA emissions are parsed out by
sectors, i.e., electrical transformation and distribution (ETD), electrical power manufacturing
(EPM), magnesium production, and electronics, while EDGAR emissions are presented as totals.
Atmosphere-based emission estimates (black lines) include uncertainties at a 95% confidence
interval (vertical black bars).

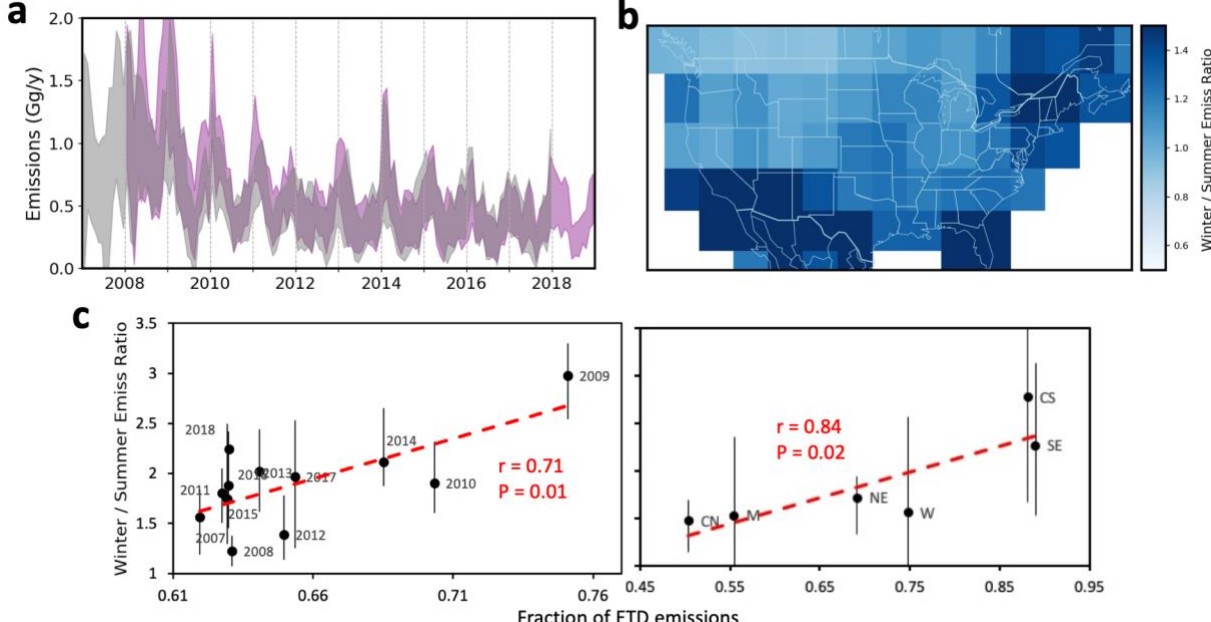

**Fig. 3**. Seasonal cycle of U.S. SF$_6$ emissions derived from atmospheric observations. (a) Monthly
emissions derived from atmospheric inversions using HYSPLIT-NAMS (in purple shading) and
WRF-STILT (in gray shading) transport simulations. The shading associated with each transport
model represents a combined uncertainty associated with 6 different inversions. (b) The winter-
to-summer emission ratios derived on a 5° × 5° grid from atmospheric observations, averaged
across all years and 12 inversion ensemble members. The winter and summer here are defined as
Nov – Feb and May – Aug. (c) Atmosphere-derived winter-to-summer emission ratios versus the
fraction of total U.S. SF$_6$ emissions from electric power transformation and distribution (ETD)
reported by EPA. Left: the ETD emission fraction versus winter-to-summer emission ratios for
annual national emissions; errorbars indicate the 2.5$^{th}$ – 97.5$^{th}$ percentile range from the 12
inversion ensembles. Right: the mean ETD emission fraction by region averaged between 2007-
2018 versus winter-to-summer emission ratios for multi-year average regional emissions over the
same period; errorbars indicate the 2.5$^{th}$ – 97.5$^{th}$ percentile range from the 12 inversion ensembles.