# Peer review of "Declining, seasonal-varying emissions of sulfur hexafluoride from the United States"

_EGUsphere, 2022_

## Referee Comment (RC1)

Declining, seasonal-varying emissions of sulfur hexafluoride from the United States point to a new mitigation opportunity

Lei Hu et. al.

**Overview**

This paper describes how emissions of SF6 have been estimated over the US using atmospheric observations and atmospheric transport modelling. These estimates have been compared to the reported inventory both at country and state level. The decline in US emissions of SF6 is seen in both the inverse modelling and the inventory, although the magnitudes differ, especially in the early years of the comparison. Through strong interaction with the US EPA several mitigation opportunities have been identified.

The paper is well-written and the results are presented very clearly. I have some questions and comments that are detailed below.

**Comments**

P3, L128: A supplement table listing the stations used and their inlet height / aircraft heights is necessary I think.

P4, L145: So in option 2 were all particles used regardless of whether they exited the US or not? It implies 500 values were used but also says it just uses particles that left the domain.

P4, L173: Particles released from the inlet height of the observation? Up to what height is the surface sensitivity modelled (surface to model BL or a fixed height)?

P4: 500 particles seems remarkably small to me, very different to the many 1000s other modelling systems use. It would be extremely helpful to understand the impacts of this decision on the US estimates. A sensitivity analysis of some sort would be very valuable, or point to a reference where this has been done and add a sentence as to the results.

P4, L180: The comparisons in the paper use EDGAR v6. Please explain the logic behind using v4.2 as the prior.

P5, L186: Are all off-diagonal elements of the transport uncertainty matrix zero? i.e. are all the observation-model uncertainties assumed to be time and space independent? For flasks every two days a long way apart physically this is okay, but not for the observations in the same vertical profile in aircraft data? How was this handled?

P5, L189: There has been no discussion on inversion grid resolutions. Uncertainty calculations from different grids - how are they combined to estimate region and US emission sigma?

P5, L194: Are cross, non-diagonal elements of the uncertainty matrix considered?

P5, L194: This formula implies that all of the inversions are independent. They are to an extent, e.g. different meteorology, but equally there are lots of dependence, e.g. the baseline options have strong correlations. The delta S term is interesting, I can see why it is added but wonder if a better idea would be to combine them all in one step. E.g. create 1000 estimates from each inversion and then estimate a mean and a std directly? I think the method described here will create uncertainties that are too small, although hard to be very sure given the delta S term.

P6, L231: How was this trend and uncertainty calculated? Do these uncertainties account for the reduction in uncertainty over time?

P18, Fig 3: Are the uncertainties using the different transport models calculated using 6 inversion each? I assume so but best to specify

P18, Fig3: Is this the inversion grid used to estimate the emissions in all cases? It would be good to explicitly show the inversion grid and to describe how it is defined.

Supp. P2, Fig S2: Why not just use 2008-2010 for both? It looks unusual that 2009-2010 are not plotted. Is it because 2009 looks so very different? This should be shown.

**Minor Text Comments**

The United States is described as 'U.S.' and 'US' - Please be consistent throughout the paper.

P2, L72: "after emissions" – remove as unnecessary or change to "after emission"

P2, L73: Add "by the U.S."

P2, L74: "accurate estimates of the magnitude and distribution of SF6 emissions"

P2, L82: I do not like the use of 'developed (ing) countries', I prefer 'Annex 1' and 'non-Annex 1' countries.

P2, L83: "(including China, one of the largest SF6 emitting countries)"

P3, L104: The same list of papers are repeated several times, maybe this should be reduced to just the key paper after they are first mentioned.

P3, L116: Manning et al 2021, ACP is a more appropriate reference as it is peer-reviewed.

P4, L150: Please re-word the sentence beginning "Because there was …back for 10 days"

P4, L155: Consider ".. to populated areas, defined as summed …". I was very unclear where the "less than 0.1 ppt (pmol ..)" came in?

P4, L 174: "which has fewer vertical levels than NAMs." – please define 'fewer'

P5, L189: I don't think the word 'ensembles' is useful here.

P5, L218: Maybe this EPA report does not need to be cited so much, it has been cited 3 times in the last two paragraphs? Or can the citation be condensed in length?

P6, L267: Reference (16)? What is this?

P7, L278: Consider replacing 'through' with 'up until'

P7, L306: 'lbs' – replace with 'mass'

P9, L410: Add the word 'reporting' e.g. 'from other reporting countries'

P11, L499: Extra line break

P12, L523: "$CO_2$" – incorrect formatting

P13, L552 and L592: "$SF_6$" – incorrect formatting

P14, L594: No year associated with Rodgers C.D.

P16, L680: Change "difference" to "different"

---

## Referee Comment (RC2)

Review of the manuscript

**Declining, seasonal-varying emissions of sulfur hexafluoride from the United States point to a new mitigation opportunity**

by Lie Hu et al.

**General Comments:**

Hu et al. present a careful top-down (TD) investigation of US SF6 emissions and their changes, based on atmospheric observations and inverse modelling, and compare these estimates with emissions reported by bottom-up (BU) inventories from the national Environmental Protection Agency (EPA) and from EDGARv6.0. A decrease of SF6 emissions as reported by the inventories is confirmed by the independent TD estimate. However, absolute values deviate significantly between BU and TD by more than a factor of two for part of the observational period. A significant seasonality in emissions is observed in the TD estimate, which is attributed to a seasonality of leakages from Electric power Transmission and Distribution (ETD) network (supposed to be high during winter at colder climate in the northern part of the US) and regular maintenance (more frequent during winter in southern parts of the US). Attributing the seasonality to the ETD source sector gives the reasoning to the title of the manuscript.

The manuscript is well written (but see my minor comments below), and fits the scope of EGUsphere. It convincingly shows that the TD approach gives important insights also into past emissions as well as into under-reporting. I see only one major shortcoming in the manuscript, which concerns the comparison of the US TD and BU emission estimates with EDGARv6.0, as presented in Fig. 1. The huge difference between the two BU inventories EDGARv6.0 and EPA of almost a factor of two around 1990 and more than a factor of four in more recent years (and also compared to the TD estimate) is rather worrying. Why is EDGARv6.0 supposedly so wrong in the US, particularly in recent years? Did the authors contact EDGAR scientists and discuss this issue with them? From my point of view, a thorough discussion on this issue needs to be added to the manuscript before it can be published.

**Specific Comments:**

*Title:*

The title creates huge expectations at the reader: Is it really a NEW opportunity to mitigate SF6? Did power companies not know that their sealings may become brittle in the cold? I would have assumed that minimizing emissions of SF6 from ETD during servicing must have been a well-known mitigation opportunity before the results of this study were available.

*Abstract:*

Lines 25-28: I suggest the sentence should read somehow like "BU *estimates* compare to TD *observations*", not the other way round (at least from an atmospheric scientist's perspective).

Line 33: … that did not or *does* not report …

Line 37: "These results …" there is no direct relation to the sentence before and I would thus change to "The results of this study demonstrate …

*Introduction:*

Line 51: … is *the* greenhouse …

Lines 75-77: You may want to cite here the early work by Maiss, M. and I. Levin (1994: Global increase of SF6 observed in the atmosphere. *Geophys. Res. Lett. 21*, 569-572) and Harnisch, J., et al. (1996: Tropospheric trends for CF4 and C2F6 since 1982 derived from SF6 dated stratospheric air; Geophys Res. Lett., 23, 1099-1102, 1996).

Lines 94-96: "This large difference *likely* stems from different input emission activity data and estimated emission factors …" Are there any other factors used to estimate BU inventories?

Lines 104-105: Which of these references describe the inverse model best? Hu et al., 2015? Then (only) this one should be cited here (to be consulted by the interested reader).

*Methods:*

Line 135: What means "US (CONUS)"?

Lines 160-161: See comment on Lines 104-105.

Line 175: What means "(Rodgers)"?

*Results and Discussion:*

Line 226: See general comment on huge deviation of EDGARv6.0 emissions from the other estimates.

Line 238: Define "GHGI".

Lines 243-245: Same comment as on Lines 25-28

Line 255: What means "GHGRP (16)"? Also Line in 267 and 269.

Line 306: "… (lbs of SF6 per …)" Think metric !

Lines 342-358: When mentioning Fig 3, please also refer to "a, b, c"

---

## Author Comment (AC1)

We appreciate two reviewers for reviewing our manuscript and bringing up their invaluable perspectives to this work. We made necessary modifications according to their suggestions. Below, the original comments are shown in black and our line-by-line responses are indicated in blue.
* * *
**Response to Reviewer #1:**

This paper describes how emissions of SF6 have been estimated over the US using atmospheric observations and atmospheric transport modelling. These estimates have been compared to the reported inventory both at country and state level. The decline in US emissions of SF6 is seen in both the inverse modelling and the inventory, although the magnitudes differ, especially in the early years of the comparison. Through strong interaction with the US EPA several mitigation opportunities have been identified.

The paper is well-written and the results are presented very clearly. I have some questions and comments that are detailed below.

**Comments**

P3, L128: A supplement table listing the stations used and their inlet height / aircraft heights is necessary I think.

Response: We provided a text file listing all the stations used, their lat, lon, alt, inlet height, and sampling date and time, measured mole fractions, and estimated background values. The link for this file is added to the manuscript.

P4, L145: So in option 2 were all particles used regardless of whether they exited the US or not? It implies 500 values were used but also says it just uses particles that left the domain.

Response: In option 2, all 500 back-trajectories were used for estimating background, no matter whether particles exited North America at the end of 10 days or not. For particles exiting before 10 days, we assign background values from the 4D empirical background at the locations where and when the particles exited the North American domain. For particles that did not exit the domain at the end of 10 days, we assign values at the locations where particles terminated at the end of 10 days. Typically, the majority of particles exited the North American domain within 10 days. But for midcontinent and eastern sites, up to 20% of particles ended up in the North American domain. This could result in some underestimate from this approach. That is why we consider our 3rd approach. This detail was now added to the manuscript.

P4, L173: Particles released from the inlet height of the observation? Up to what height is the surface sensitivity modelled (surface to model BL or a fixed height)?

Response: Particles were released at the sampling inlet heights. Surface sensitivity was modelled by integrating the influence of particles between the modeled surface to modeled boundary layer. The surface sensitivity was directly calculated as HYSPLIT and STILT model outputs, following the method described in Lin et al. (2003). This is now added to the method section.

Lin, J. C., Gerbig, C., Wofsy, S. C., Andrews, A. E., Daube, B. C., Davis, K. J., and Grainger, C. A. (2003), A near-field tool for simulating the upstream influence of atmospheric observations: The Stochastic Time-Inverted Lagrangian Transport (STILT) model, *J. Geophys. Res.*, 108, 4493, doi:10.1029/2002JD003161, D16.

P4: 500 particles seems remarkably small to me, very different to the many 1000s other modelling systems use. It would be extremely helpful to understand the impacts of this decision on the US

estimates. A sensitivity analysis of some sort would be very valuable, or point to a reference where this has been done and add a sentence as to the results.

Response: The impact of particle numbers on simulated footprints were tested in the early development of the STILT model and described in Gerbig et al. (2003). HYSPLIT uses similar algorithms as STILT to calculate footprints. Gerbig et al. (2003) tested particle numbers between 100 – 1000. Because of the stochastic nature of the STILT model, the error resulting from using the 100 particles is only 13%. We also tested using 500 versus 1000 particles in the early development of our inverse method in 2012. We did not find significant differences in our generated footprints. Furthermore, some early North American inversion studies published at that time also used 500 particles in STILT footprint simulations (Miller et al., 2013). Given these early findings and the exponential increase of computation cost in running STILT and HYSPLIT models, we chose to use 500 particles in our STITL and HYSPLIT simulations, which was also used in our previous inversion studies for other trace gases (Hu et al., 2015, 2016, 2017, 2019, 2021). In the revision, we added some of the references mentioned here.

Gerbig, C., Lin, J. C., Wofsy, S. C., Daube, B. C., Andrews, A. E., Stephens, B. B., Bakwin, P. S., and Grainger, C. A.: Toward constraining regional-scale fluxes of CO2 with atmospheric observations over a continent: 2. Analysis of COBRA data using a receptor-oriented framework, Journal of Geophysical Research: Atmospheres, 108, https://doi.org/10.1029/2003JD003770, 2003.

Miller, S. M., Wofsy, S. C., Michalak, A. M., Kort, E. A., Andrews, A. E., Biraud, S. C., Dlugokencky, E. J., Eluszkiewicz, J., Fischer, M. L., Janssens-Maenhout, G., Miller, B. R., Miller, J. B., Montzka, S. A., Nehrkorn, T., and Sweeney, C.: Anthropogenic emissions of methane in the United States, Proceedings of the National Academy of Sciences, 10.1073/pnas.1314392110, 2013.

P4, L180: The comparisons in the paper use EDGAR v6. Please explain the logic behind using v4.2 as the prior.

Response: EDGARv6 only provides national total emission estimates of SF6. No gridded products were available. That is why we used gridded emissions from EDGARv4.2. But we just found out a new version EDGARv7 just came out online during the time period when we were trying to address this comment. EDGARv7 has products on national total and gridded emissions. We compared its national totals and regional distributions with EDGARv6 and EDGARv4.2, the national totals are identical with EDGARv6 for the overlapped years and the gridded products has similar distributions as EDGARv4.2. So, we updated figures and discussions in the paper with EDGARv7. The general discussion still holds with EDGARv7. We did not run inversions with the gridded EDGARv7 as a priori. This is because we tried inversions with multiple priors with different magnitudes and distributions earlier. Our regional and national total emission estimates were fairly insensitive to the prior. So, we don't expect significant changes in our results, even if we try to use EDGARv7 as our prior.

P5, L186: Are all off-diagonal elements of the transport uncertainty matrix zero? i.e. are all the observation-model uncertainties assumed to be time and space independent? For flasks every two days a long way apart physically this is okay, but not for the observations in the same vertical profile in aircraft data? How was this handled?

Response: Our off-diagonal elements in the model-data mismatch covariance matrix were 0, assuming all observations we included were independent. In each aircraft profile, it generally collects two flasks >500 meters apart below 1000 meters and then one flask every km for samples collected at 1 km and above. We assumed samples taken from individual aircraft profiles are independent, as each of them represents slightly different footprints. This assumption is likely not

optimal, and it is likely free tropospheric observations from aircraft profiles may be correlated. But the impact of this assumption on the overall error characterization is likely minimal. This is because (1) the sensitivity of free tropospheric observations on surface fluxes is relatively small. They don't have much influence on as the derived fluxes and flux errors. (2) The errors from our inversion are dominated by the spread of inversion ensemble members rather than posterior errors from each inversion.

P5, L189: There has been no discussion on inversion grid resolutions. Uncertainty calculations from different grids - how are they combined to estimate region and US emission sigma?
Response: As we stated in the paper, we followed the same methodology as our previous setup that was described in detail in our previous publications (Hu et al., 2015; Hu et al., 2017). Our inversion grid is $1^o$ x $1^o$ at weekly time steps. Derived posterior emissions and uncertainties were aggregated at regional and national scales at monthly time step. When we aggregate them for uncertainties, we consider the temporal and spatial correlations in the non-diagonal grid cells in our posterior uncertainty matrix. This detail was now added into the paper.

P5, L194: Are cross, non-diagonal elements of the uncertainty matrix considered?
Response: yes, the cross, non-diagonal elements of uncertainty matrix was considered in the posterior uncertainty calculation in each run. This detail was now added into the manuscript.

P5, L194: This formula implies that all of the inversions are independent. They are to an extent, e.g. different meteorology, but equally there are lots of dependence, e.g. the baseline options have strong correlations. The delta S term is interesting, I can see why it is added but wonder if a better idea would be to combine them all in one step. E.g. create 1000 estimates from each inversion and then estimate a mean and a std directly? I think the method described here will create uncertainties that are too small, although hard to be very sure given the delta S term.
Response: The formula (Eq. 1) actually implies all 12 inversions are correlated, as the first term is an average uncertainty from 12 inversions. When all 12 inversions are independent, this term would need to be divided by $\sqrt{12}$ and hence the uncertainty would be even smaller than we estimated. No change was made here.

P6, L231: How was this trend and uncertainty calculated? Do these uncertainties account for the reduction in uncertainty over time?
Response: the annual uncertainties are strictly calculated from Eq. 1 and then divided by the square root of the number of years to get the uncertainty for time periods of interest. This calculation did account for the reduction in uncertainty over time. No change was made here.

P18, Fig 3: Are the uncertainties using the different transport models calculated using 6 inversion each? I assume so but best to specify
Response: Yes, and we have now added the specification indicating each color shading represents uncertainties from 6 inversions.

P18, Fig3: Is this the inversion grid used to estimate the emissions in all cases? It would be good to explicitly show the inversion grid and to describe how it is defined.
Response: Our inversions were performed at $1^o \times 1^o$ resolution (this detail is added to the method section now). Because the uncertainties at $1^o \times 1^o$ are large and results on this spatial scale tend to be noisier, so we generally don't discuss the grid-scale results. But here, to show the spatial distribution, we coarsened results to $5^o \times 5^o$, where uncertainties should be reduced relative to $1^o \times 1^o$.

Supp. P2, Fig S2: Why not just use 2008-2010 for both? It looks unusual that 2009-2010 are not plotted. Is it because 2009 looks so very different? This should be shown.
Response: The reason to show 2007 – 2008 and 2011 – 2018 is because they correspond to periods with maximum and minimum emissions. Plotting their differences allow us to see the strongest declining signal.  Adding 2009 – 2010 when emissions were declining dilute this signal, but does not alter the conclusion.  Given this comment, we have change the time periods to 2008 and 2011- 2017, so that maps were shown for the same time periods for both transport models.

**Minor Text Comments**
The United States is described as 'U.S.' and 'US' - Please be consistent throughout the paper.
Response: We corrected "US" to "U.S." throughout the paper.

P2, L72: "after emissions" – remove as unnecessary or change to "after emission"
Response: "after emissions" were removed.

P2, L73: Add "by the U.S."
Response: added.

P2, L74: "accurate estimates of the magnitude and distribution of SF6 emissions"
Response: accepted.

P2, L82: I do not like the use of 'developed (ing) countries', I prefer 'Annex 1' and 'non-Annex 1' countries.
Response: We changed it to Annex I (mostly developed) countries here.

P2, L83: "(including China, one of the largest SF6 emitting countries)"
Response: change was made.

P3, L104: The same list of papers are repeated several times, maybe this should be reduced to just the key paper after they are first mentioned.
Response: accepted. Extra citations were removed.

P3, L116: Manning et al 2021, ACP is a more appropriate reference as it is peer-reviewed.
Response: we added the recommended citation and removed Manning et al. (2017).

P4, L150: Please re-word the sentence beginning "Because there was ...back for 10 days"
Response: we reworded this sentence.

P4, L155: Consider ".. to populated areas, defined as summed ...". I was very unclear where the "less than 0.1 ppt (pmol ..)" came in?
Response: Please refer to our previous paper on why we chose this threshold. Since the main point is to compare the estimated background with observations that have low surface sensitivity, we removed the second half of the sentence to avoid similar confusion for other readers.

P4, L 174: "which has fewer vertical levels than NAMs." – please define 'fewer'
Response: We changed it to "which only has 26 vertical levels".

P5, L189: I don't think the word 'ensembles' is useful here.

Response: We deleted "ensembles" here.

P5, L218: Maybe this EPA report does not need to be cited so much, it has been cited 3 times in the last two paragraphs? Or can the citation be condensed in length?
Response: We removed the second citation on the EPA inventory in this paragraph and condensed its length.

P6, L267: Reference (16)? What is this?
Response: "(16)" is deleted.

P7, L278: Consider replacing 'through' with 'up until'
Response: we replaced it with "up until".

P7, L306: 'lbs' – replace with 'mass'
Response: replaced with 'kg'.

P9, L410: Add the word 'reporting' e.g. 'from other reporting countries'
Response: added.

P11, L499: Extra line break
Response: Removed the extra break here.

P12, L523: "CO2" – incorrect formatting
Response: Corrected.

P13, L552 and L592: "SF6" – incorrect formatting
Response: Corrected.

P14, L594: No year associated with Rodgers C.D.
Response: added "2000"

P16, L680: Change "difference" to "different"
Response: Corrected.
* * *
**Response to reviewer #2:**
  Review of the manuscript

  **Declining, seasonal-varying emissions of sulfur hexafluoride from the United States point to a new mitigation opportunity**
  by Lei Hu et al.

  **General Comments:**

  Hu et al. present a careful top-down {TD} investigation of US SF6 emissions and their changes, based on atmospheric observations and inverse modelling, and compare these estimates with emissions reported by bottom-up {BU} inventories from the national Environmental Protection Agency {EPA} and from EDGARv6.0. A decrease of SF6 emissions as reported by the inventories is confirmed by the independent TD estimate. However, absolute values deviate significantly between BU and

TD by more than a factor of two for part of the observational period. A significant seasonality in emissions is observed in the TD estimate, which is attributed to a seasonality of leakages from Electric power Transmission and Distribution {ETD} network {supposed to be high during winter at colder climate in the northern part of the US} and regular maintenance {more frequent during winter in southern parts of the US}. Attributing the seasonality to the ETD source sector gives the reasoning to the title of the manuscript.

The manuscript is well written {but see my minor comments below}, and fits the scope of EGUsphere. It convincingly shows that the TD approach gives important insights also into past emissions as well as into under-reporting. I see only one major shortcoming in the manuscript, which concerns the comparison of the US TD and BU emission estimates with EDGARv6.0, as presented in Fig. 1. The huge difference between the two BU inventories EDGARv6.0 and EPA of almost a factor of two around 1990 and more than a factor of four in more recent years {and also compared to the TD estimate} is rather worrying. Why is EDGARv6.0 supposedly so wrong in the US, particularly in recent years? Did the authors contact EDGAR scientists and discuss this issue with them? From my point of view, a thorough discussion on this issue needs to be added to the manuscript before it can be published.

Response: Thanks to the reviewer for the useful comments on our manuscript. We appreciate that the reviewer believes that our top-down results provide important insights on US $SF_6$ emissions.

We agree with the reviewer that the substantial difference in inventory results (EDGAR vs USEPA) and between EDGAR and our new top-down estimates is concerning. But an understanding of this issue will necessarily be an involved process with many other scientists that is well beyond the scope of this paper. We think it entirely appropriate to identify the differences here in this work so that future analyses can lead to an understanding of why they exist. We have attempted to start the discussion between scientists responsible for creating the EDGAR inventory and look forward to future collaboration and interaction with them so that our understanding of these differences can be improved. As a first step, we added to the revision a supplemental figure and discussion on how the EDGAR inventory is different from the EPA inventory and our top-down estimates by sector and region. This analysis points to key sectors and regions where the differences are the largest and this will help guide the conversation with the EDGAR community once that relationship matures.

*Title:*

The title creates huge expectations at the reader: Is it really a NEW opportunity to mitigate SF6? Did power companies not know that their sealings may become brittle in the cold? I would have assumed that minimizing emissions of SF6 from ETD during servicing must have been a well-known mitigation opportunity before the results of this study were available.

Response: Undoubtedly, they did know of this opportunity but it hadn't been well quantified until now. To simplify, we changed our title to "Declining, seasonal-varying emissions of sulfur hexafluoride from the United States"

*Abstract:*

Lines 25-28: I suggest the sentence should read somehow like "BU *estimates* compare to TD *observations*", not the other way round {at least from an atmospheric scientist's perspective}.

Response: A clearer rephrasing proposed in main document.

Line 33: . that did not or *does* not report .
Response: corrected "do" to "does" here.

Line 37: "These results ." there is no direct relation to the sentence before and I would thus change to "The results of this study demonstrate .
Response: made the suggested change.

Line 51: . is the greenhouse .
Response: made the suggested change.

Lines 75-77: You may want to cite here the early work by Maiss, M. and I. Levin {1994: Global increase of SF6 observed in the atmosphere. *Geophys. Res. Lett. 21*, 569-572} and Harnisch, J., et al. {1996: Tropospheric trends for CF4 and C2F6 since 1982 derived from SF6 dated stratospheric air; Geophys Res. Lett., 23, 1099-1102, 1996}.

Response: The suggested references were added.

Lines 94-96: "This large difference *likely* stems from different input emission activity data and estimated emission factors ." Are there any other factors used to estimate BU inventories?
Response: The cause for the large difference is not fully known yet. But we did compare EDGAR with EPA GHGI. We found the difference is primarily from the ETD sector. Correspondingly, we made some changes here to reflect this point.

Lines 104-105: Which of these references describe the inverse model best? Hu et al., 2015? Then {only} this one should be cited here {to be consulted by the interested reader}.
Response: We removed some extra citations and kept the ones that are most relevant. We do think it is useful to have readers aware the same modeling method has been applied to other gases previously.

*Methods:*

Line 135: What means "US {CONUS}"?
Response: CONUS represents the contiguous US.

Lines 160-161: See comment on Lines 104-105.
Response: We deleted extra references here.

Line 175: What means "{Rodgers}"?
Response: It should be Rodgers (2000). It is now corrected.

*Results and Discussion:*

Line 226: See general comment on huge deviation of EDGARv6.0 emissions from the other estimates.
Response: see our response above.

Line 238: Define "GHGI".
Response: added the definition.

Lines 243-245: Same comment as on Lines 25-28
Response: reworded.

Line 255: What means "GHGRP {16}"? Also Line in 267 and 269.
Response: deleted "(16)" in both places.

Line 306: " _ {lbs of SF6 per _ }" Think metric !
Response: changed "lbs" to "kg"

Lines 342-358: When mentioning Fig 3, please also refer to "a, b, c"
Response: added a, b, c to places, when mentioning Fig. 3.